# Pragmatic, Prospective Comparative Effectiveness Trial of Carbon Ion Therapy, Surgery, and Proton Therapy for the Management of Pelvic Sarcomas (Soft Tissue/Bone) Involving the Bone: The PROSPER Study Rationale and Design

**DOI:** 10.3390/cancers15061660

**Published:** 2023-03-08

**Authors:** Bradford S. Hoppe, Ivy A. Petersen, Benjamin K. Wilke, Todd A. DeWees, Reiko Imai, Eugen B. Hug, Maria Rosaria Fiore, Jürgen Debus, Piero Fossati, Shigeru Yamada, Ester Orlandi, Qing Zhang, Cihang Bao, Katharina Seidensaal, Byron C. May, Anna C. Harrell, Matthew T. Houdek, Laura A. Vallow, Peter S. Rose, Michael G. Haddock, Jonathan B. Ashman, Krista A. Goulding, Steven Attia, Sunil Krishnan, Anita Mahajan, Robert L. Foote, Nadia N. Laack, Sameer R. Keole, Chris J. Beltran, Eric M. Welch, Mohammed Karim, Safia K. Ahmed

**Affiliations:** 1Department of Radiation Oncology, Mayo Clinic, Jacksonville, FL 32224, USA; 2Department of Radiation Oncology, Mayo Clinic, Rochester, MN 55905, USA; 3Department of Orthopedic Surgery, Mayo Clinic, Jacksonville, FL 32224, USA; 4Division of Clinical Trials and Biostatistics, Mayo Clinic, Phoenix, AZ 85054, USA; 5Department of Radiation Oncology, Mayo Clinic, Phoenix, AZ 85054, USA; 6Division of Radiation Oncology, QST Hospital, National Institutes for Quantum and Radiological Science and Technology, Chiba 263-8555, Japan; 7Department of Radiation Oncology, MedAustron Ion Therapy Center, 2700 Wiener Neustadt, Austria; 8Radiation Oncology Clinical Department, National Center for Oncological Hadrontherapy (CNAO), 27100 Pavia, Italy; 9Department of Radiation Oncology, University Hospital Heidelberg, 69120 Heidelberg, Germany; 10Clinical Cooperation Unit Radiation, German Cancer Research Center, 69120 Heidelberg, Germany; 11Department for Basic and Translational Oncology and Hematology, Karl Landsteiner University of Health Sciences, 3500 Krems, Austria; 12Department of Radiation Oncology, Shanghai Proton and Heavy Ion Center, Fudan University Cancer Hospital, Shanghai 201102, China; 13Department of Orthopedic Surgery, Mayo Clinic, Rochester, MN 55905, USA; 14Department of Orthopedic Surgery, Mayo Clinic, Phoenix, AZ 85054, USA; 15Division of Hematology and Medical Oncology, Mayo Clinic, Jacksonville, FL 32224, USA; 16Department of Radiation Oncology, University of Texas Health Houston Neurosciences-Texas Medical Center, Houston, TX 77030, USA

**Keywords:** cancer treatment, carbon ion, clinical trial, pelvic sarcoma, proton therapy, radiotherapy, surgical treatment

## Abstract

**Simple Summary:**

Surgical treatment of pelvic sarcoma involving the bone is the standard of care but is associated with several treatment sequelae and reduced functional quality of life. Treatment with photon and proton radiotherapy is associated with relapse. Carbon ion radiotherapy may reduce both relapse rates and treatment sequelae. The PROSPER study is a tricontinental, nonrandomized, prospective, three-arm, pragmatic trial evaluating carbon ion radiotherapy, proton therapy, and the surgical treatment of pelvic sarcoma involving the bone. We describe the rationale and design of this clinical trial currently open to enrollment.

**Abstract:**

Surgical treatment of pelvic sarcoma involving the bone is the standard of care but is associated with several sequelae and reduced functional quality of life (QOL). Treatment with photon and proton radiotherapy is associated with relapse. Carbon ion radiotherapy (CIRT) may reduce both relapse rates and treatment sequelae. The PROSPER study is a tricontinental, nonrandomized, prospective, three-arm, pragmatic trial evaluating treatments of pelvic sarcoma involving the bone. Patients aged at least 15 years are eligible for inclusion. Participants must have an Eastern Cooperative Oncology Group Performance Status score of two or less, newly diagnosed disease, and histopathologic confirmation of pelvic chordoma, chondrosarcoma, osteosarcoma, Ewing sarcoma with bone involvement, rhabdomyosarcoma (RMS) with bone involvement, or non-RMS soft tissue sarcoma with bone involvement. Treatment arms include (1) CIRT (*n* = 30) delivered in Europe and Asia, (2) surgical treatment with or without adjuvant radiotherapy (*n* = 30), and (3) proton therapy (*n* = 30). Arms two and three will be conducted at Mayo Clinic campuses in Arizona, Florida, and Minnesota. The primary end point is to compare the 1-year change in functional QOL between CIRT and surgical treatment. Additional comparisons among the three arms will be made between treatment sequelae, local control, and other QOL measures.

## 1. Introduction

An estimated 3910 cases of bone cancer are diagnosed annually in the US, constituting 0.2% of all newly diagnosed cancer cases [1]. These neoplasms include chordoma, chondrosarcoma, Ewing sarcoma, and osteosarcomas, among others. Up to 14% of all malignant bone tumors are located in the pelvic bones [2]. Multimodal treatment is always required and should be performed at a center with expertise in treating sarcoma [3]. Local treatment may include surgery, radiotherapy, or both, depending on the tumor location, histopathologic characteristics, anticipated surgical resection issues, anticipated surgical sequelae, and patient preferences. 

Many oncologists consider surgical treatment to be the optimal choice for the local treatment of pelvic bone sarcoma because surgical treatment yields higher local control rates than does definitive radiotherapy [4,5,6,7]. For example, Houdek et al [5] reported a 5-year local control rate of 80% with surgical resection for sacrococcygeal chordoma, as compared with a 70% to 75% local control rate with definitive radiotherapy. Similarly, the Children’s Oncology Group reported a six-fold higher local recurrence rate for pelvic Ewing sarcoma treated with definitive radiotherapy than with surgical treatment alone [8]. Additionally, the Cooperative Osteosarcoma Study Group reported that the omission of surgical treatment was correlated with a high risk of relapse among a cohort of patients with pelvic osteosarcoma [9]. Despite these findings, each form of local therapy poses substantial challenges with regard to disease outcomes, as well as late adverse events, reduced quality of life (QOL), and poor functional outcomes.

For resectable pelvic bone and soft tissue sarcoma, surgical treatment may require extensive resection and reconstruction. Surgical treatment sequelae are more likely to occur when the tumor involves multiple pelvic bones, the lumbosacral plexus, and/or the pelvic vasculature [10]. Additionally, the preservation of lower extremity, bowel, and bladder function often requires accepting close surgical margins. Therefore, surgical treatment may negatively affect patient QOL and function [11,12]. 

Definitive radiotherapy is used as an alternative to surgical treatment for pelvic bone sarcoma when resection is not feasible or would result in severe treatment sequelae and/or when the patient is not a candidate or is unwilling to undergo surgical treatment [4]. However, high radiotherapy doses are needed for local control in this setting because of the low α/β ratio for most of these bone tumors [13]. National Comprehensive Cancer Network (NCCN) guidelines recommend doses greater than 70 Gy for bone sarcoma, which exceeds the tolerance of surrounding healthy pelvic tissues, such as the small bowel and lumbosacral plexus. For this reason, the NCCN recommends the use of proton beam therapy (PT) or carbon ion radiotherapy (CIRT) for treating pelvic bone sarcoma [4]. 

With conventional photon radiotherapy, the maximum radiation dose is deposited after entering the body and is attenuated as it passes through the entire body. This dose distribution limits the use of the most effective radiotherapy doses and exposes a considerable amount of healthy tissue to the harmful effects of radiotherapy. PT uses positively charged proton ions and delivers a maximum dose to a specific depth without an exit dose because of the physical characteristics of the Bragg peak. Accordingly, PT delivers a high radiotherapy dose to the treatment volume while exposing healthy tissues beyond the target volume to a lower dose than does conventional photon radiotherapy. This PT dose distribution also permits a safer dose escalation and/or hypofractionation. Many studies have reported that PT is safe and effective for the treatment of pelvic sarcoma [14,15,16,17]. 

CIRT uses charged carbon ions that are heavier than proton ions. CIRT also has the physical advantage of the characteristic Bragg peak, which allows for the depth of the carbon ion beam to be modulated and protection of healthy tissues around the treatment volume. Moreover, carbon ions have high linear energy transfer (LET) and a low oxygen enhancement ratio, with high relative biological effectiveness (RBE) that results in clustered DNA damage that is difficult to repair and that causes extensive tumor cell death [18,19]. In addition, CIRT can effectively enhance apoptosis in radioresistant cancer stem cells, which have been characterized in bone sarcomas. These radiobiological characteristics of CIRT appear to improve local control outcomes of radioresistant pelvic bone sarcomas such as chordoma, osteosarcoma, and chondrosarcoma [20,21,22,23,24,25,26]. 

Currently, only a few international institutions use CIRT to treat pelvic bone sarcoma [27]. Accordingly, data evaluating the potential differences in oncologic outcomes, treatment sequelae, and functional outcomes between CIRT and surgical treatment or PT are scarce. Recently, Yolcu et al [28] compared outcome data from registries of patients with sacral chordoma treated with the following: (1) CIRT at the National Institute for Quantum and Radiological Science and Technology (QST Hospital) in Chiba, Japan; (2) en bloc surgical resection at Mayo Clinic in Rochester, MN, USA; (3) en bloc surgical resection with or without photon therapy at various institutions reporting to the National Cancer Database (NCDB). Overall survival rates were worse with photon therapy, as reported by the NCDB, than with CIRT at QST Hospital. Additionally, the overall survival rate was similar for patients treated with CIRT and those treated with surgical resection (with or without radiotherapy), as reported by the NCDB. When comparing the outcomes of patients treated at QST Hospital with those treated with en bloc surgical resection at Mayo Clinic, no significant difference in local control or overall survival was observed. However, patients who underwent CIRT had less peripheral motor neuropathy and marginally better functional outcomes. Similar outcomes were reported by Outani et al [12]. 

Both of these studies were limited by a retrospective analysis of different combinations of patient registry data, with only physician-reported outcomes. The SACRO trial (NCT02986516), an international, multicenter, observational study of localized sacral chordoma, includes both randomized arms of surgical treatment vs. high-dose definitive radiotherapy (CIRT, PT, and mixed photon-PT), as well as two nonrandomized prospective arms. In the nonrandomized arms, surgery or PT is selected according to patient and physician preference. This trial is ongoing, and accrual has occurred as planned for the nonrandomized arms but has been difficult for the randomized arms. For this reason, we developed the PROSPER study (NCT05033288). This trial is a pragmatic, prospective, multicenter study comparing functional outcomes, treatment sequelae (reported by both patients and physicians), and local control for patients with pelvic bone sarcoma who undergo surgical treatment, PT, or CIRT.

## 2. Research Methods and Analysis

The purpose of the PROSPER study (ClinicalTrials.gov, NCT05033288) was to compare patient-reported outcomes, adverse events/treatment sequelae, and local tumor control for bone/soft tissue sarcoma involving the pelvic bone, among three treatment arms: (1) CIRT, (2) surgical treatment, and (3) PT (Figure 1). 

The primary aim of the study is to compare patient-reported health-related QOL outcomes and treatment sequelae after CIRT with those after surgical treatment. We acknowledge that the CIRT and surgical treatment arms will have considerably different disease-specific characteristics (e.g., tumor stage, size, and extent), with a disproportionate number of patients who undergo CIRT having larger and higher-stage tumors involving the sacral plexus. However, if the functional outcomes and treatment sequelae data show an advantage for CIRT, despite these differences, then CIRT would most likely yield the same, or even better, favorable functional outcomes when comparing the two treatment arms containing disease severity-matched patient cohorts.

The secondary aim of the study is to determine whether CIRT improves local control in comparison with PT. This aim was chosen because CIRT is considered to be more biologically effective than radiotherapies with a lower LET (i.e., photon therapy or PT) for treating radioresistant tumors and may reduce the risk of local recurrence. 

The PROSPER study was designed as a prospective, parallel cohort study. Although a randomized study would provide a higher level of evidence, several feasibility and administrative challenges would arise with a randomized trial. Specifically, randomization between the surgical treatment and definitive treatment with the CIRT or PT arms would have been problematic. Even the ongoing SACRO trial, which has enrolled more than 100 patients, could not randomly assign patients, and all patients were treated in the prospective cohorts. 

In the PROSPER study, patients with unresectable tumors will be enrolled in the trial, although they will not be eligible to be randomly assigned to the surgical treatment arm. Although randomization between the CIRT and PT treatment arms would potentially be possible, the logistics of randomizing these arms would still pose challenges. The PROSPER study protocol was activated at all 3 Mayo sites in 2022: at Mayo Clinic in Florida on 20 January, Mayo Clinic in Arizona on 14 February, and Mayo Clinic in Minnesota on 1 June. The participating CIRT sites are currently preparing to activate the trial. To date, no patients have been enrolled in the PROSPER study. We anticipate that the enrollment period will require up to 36 months for accrual. 

Mayo Clinic will enroll most of the patients in the trial but will not have the ability to treat patients with CIRT until 2027, when Mayo Clinic in Florida will open its CIRT center to patients. Although Mayo Clinic investigators could send patients to Europe or Asia to receive CIRT, treatment costs and travel restrictions would be challenging because the reimbursement of costs associated with sending patients abroad is currently unresolved for CIRT. Indeed, during the development of the PROSPER study, a randomized trial led by the University of Texas Southwestern Medical Center assessing CIRT vs. intensity-modulated radiotherapy/PT treatment of patients with pancreatic cancer (NCT03536182) was opened, closed, and withdrawn when the COVID-19 pandemic restricted travel, as well as the ability to randomly assign US patients to centers with CIRT in Europe and Asia.

Because of the low incidence of bone/soft tissue sarcoma involving the pelvis, the PROSPER study was designed to be pragmatic, with generous eligibility criteria to avoid ineligibility (except for critical factors) and encourage collaboration among several CIRT centers. The CIRT centers participating in the PROSPER study use different RBE dose calculation models, dose fractionation schemes, and target volumes. We concluded that allowing these variations in the study was important to engage various international CIRT centers in one of the first multicontinental CIRT studies [29]. Although permitting such CIRT variability in our study will yield a heterogeneous cohort in the CIRT treatment arm that could obscure significant outcomes, we will scrutinize the results according to each treatment plan, which will be centralized, and evaluate the patterns associated with different functional outcomes, adverse events/treatment sequelae, and rates of local recurrence.

Inclusion criteria for enrollment in the PROSPER study are as follows: male and female patients aged at least 15 years, with an Eastern Cooperative Oncology Group Performance Status score of 2 or less, newly diagnosed disease, and histopathologic confirmation of pelvic chordoma, chondrosarcoma, osteosarcoma, Ewing sarcoma with bone involvement, rhabdomyosarcoma with bone involvement, or nonrhabdomyosarcoma soft tissue sarcoma with bone involvement. Patients will be required to provide written informed consent and complete questionnaires by themselves, or with assistance, and must agree to use adequate contraception if of childbearing age. Patients will be excluded if they have distant metastatic disease, as evidenced by clinical examination or imaging, or if they are receiving palliative-intent treatment, have recurrent disease, or have received prior radiotherapy to the tumor site or the surrounding area that would cause an overlap of radiotherapy doses in healthy tissues.

CIRT facilities in Europe and Asia will enroll patients eligible for treatment with curative-intent CIRT in treatment arm 1. Mayo Clinic campuses in Arizona, Florida, and Minnesota will enroll patients in treatment arm 2 if they are examined in the Mayo Clinic orthopedic oncology or neurosurgery departments and deemed by a multidisciplinary team to be eligible to undergo definitive surgical treatment. Mayo Clinic (all campuses) will enroll patients in treatment arm 3 if they are examined in the Mayo Clinic radiation oncology department and deemed by a multidisciplinary team to be eligible to receive definitive treatment with PT.

### 2.1. Treatment Arm 1

Five CIRT centers participated in the development of the PROSPER study, including QST Hospital (Chiba, Japan), Shanghai Proton and Heavy Ion Center (Shanghai, China), Italian National Center for Oncological Hadrontherapy (Pavia, Italy), MedAustron (Wiener Neustadt, Austria), and the Heidelberg Ion Beam Therapy Center (Heidelberg, Germany). For all sites, CIRT is a standard-of-care treatment for unresectable sarcoma tumors involving the bone(s) of the pelvis. In accordance with the pragmatic approach of the PROSPER study, no strict guidelines were developed to specify target delineation, treatment planning, RBE dose calculation model, total dose, or dose fractionation. Each site has developed its own method for managing these tumors, which was a requirement for multicenter collaboration. However, Digital Imaging and Communications in Medicine (DICOM)-standardized treatment plans will be centralized for review, including variations in target selection and delineation, target coverage, organ at-risk (OAR) dose constraints, total dose, and dose fractionation. Adjuvant systemic chemotherapy is allowed according to institutional policies. 

### 2.2. Treatment Arm 2

Surgical treatment will be performed at all 3 Mayo Clinic campuses (Phoenix, AZ, USA; Jacksonville, FL, USA; Rochester, MN, USA). The specific type of surgical procedure will be at the discretion of the surgeon for each patient. Adjuvant radiotherapy and/or chemotherapy will be allowed according to institutional approaches to treatment. 

### 2.3. Treatment Arm 3

PT will be administered at all 3 Mayo Clinic campuses. Target delineation, treatment planning, total dose, OAR dose constraints, and dose fractionation will be determined by the treating radiation oncologist, but physicians are expected to use previously agreed-upon health system treatment approaches.

### 2.4. Outcomes Measures

All patients will complete Patient-Reported Outcomes Measurement Information System-29 (PROMIS-29) and European Organisation for Research and Treatment of Cancer (EORTC) colorectal cancer–specific quality of life (QLQ-CR29) questionnaires at baseline and posttreatment time points.

Baseline patient-, disease-, and treatment-specific details will be collected and entered into our Research Electronic Data Capture (REDCap) system. After treatment is completed, the full radiation data set, including initial and any adaptive modifications to the data set (DICOM RT_PLAN, RT_STRUCT, RT_IMAGE, RT_DOSE), will be anonymized (except for the patient study number) and transferred to the Mayo Clinic DICOM database. Dose–volume histograms (DVHs) will be provided for the following OARs: the small bowel, large bowel, cauda equina, sacral plexus [30], femoral heads, rectum, and bladder. Oncologic outcomes and adverse events will also be entered in our REDCap system.

### 2.5. Follow-Up

Standard follow-up will include medical history and physical examination with Eastern Cooperative Oncology Group Performance Status scores at approximately 3, 6, 12, 24, 36, 48, and 60 months. Magnetic resonance imaging of the pelvis will be preferred, but computed tomography or positron emission tomography/computed tomography will be acceptable if the patient is unable to undergo magnetic resonance imaging. Imaging findings, treatment sequelae, and PROMIS-29 and EORTC QLQ-CR29 responses will be collected at each follow-up examination.

Assessments of treatment sequelae will include the following Common Terminology Criteria for Adverse Events v5.0: gastrointestinal tract (diarrhea, nausea, vomiting, perforation, stenosis, ulcer, colitis, obstruction, fistula, fecal incontinence, and necrosis), urinary tract (dysuria, fistula, incontinence, urgency, and retention), musculoskeletal (osteonecrosis, soft tissue necrosis, fracture, muscle weakness, lower limb, extremity pain, pelvic pain, lymphedema, soft tissue fibrosis, and gait disturbance), sexual function (dyspareunia in women and erectile dysfunction in men), and nervous system (motor neuropathy, sensory neuropathy, phantom pain, and radiculitis) disorders.

### 2.6. Statistical Consideration

For our primary aim of comparing patient-reported QOL outcomes and treatment sequelae after CIRT with those after surgical treatment, we will use the functional domain of the PROMIS-29 questionnaire. We will specifically compare the mean difference in the functional QOL from before treatment (baseline) to 1 year after the end of treatment, between treatment arms 1 and 2, with a one-sided, two-sample *t* test of independent means. On the basis of previous studies and retrospective clinical data, we conservatively expect a mean decrease in functional QOL of 2 points for treatment arm 1 and 4.6 points for treatment arm 2 (higher than the defined minimal clinically important difference), with an SD of 4 points in both arms. We will obtain 80% power (α = 0.05) to detect significantly improved functional QOL between the 2 treatment arms with 30 patients in each arm (*N* = 90 patients for all 3 arms).

To reduce the likelihood of underpowering the trial because of screening failures, loss to follow-up, death, and other causes that could lower the number of patients reaching the 5-year follow-up study end point, we will recruit an additional 6 patients per treatment arm (*n* = 36 patients per arm, *N* = 108).

Data will be stored and maintained in the external Mayo Clinic REDCap system. Analyses for the primary aim will be completed when all eligible patients have completed the PROMIS-29 questionnaire at baseline and 1 year after the end of treatment. The PROMIS-29 functional score will be calculated, and median, mean, and 95% CI values will be compared between each treatment arm, with one-sided, two-sample *t* tests. Analyses for the secondary aim of determining whether CIRT improves local control in comparison with PT will be completed after the last treated patient has been followed for 3 years. Analysis will be completed for all patients; however, because of the possibility of heterogeneous outcomes resulting from differences in histopathologic findings, analyses of local control will be stratified according to histopathologic characteristics. Subset analyses will also be performed for sacral chordoma vs. non-sacral chordoma. The proportion of patients with local control at 3 years and 95% CI in treatment arms 1 and 3 will be calculated with a one-sided noninferiority test. 

The Kaplan–Meier method with likelihood ratio tests will be used to evaluate all clinical outcomes, which will include local recurrence, regional progression, distant metastases, progression-free survival, and overall survival, for each treatment arm. Subset and stratified analyses according to histopathologic characteristics will also be conducted for all time-to-event analyses. 

Exploratory analyses of local control between arm 2 and arms 1 and 3, treatment sequelae, and DVH data will be performed with standard logistic regression analysis of acute- (<6 months) and late-onset (≥6 months) treatment sequelae. Area under the receiver operating characteristic curve analyses will be performed for each DVH measure to determine the effect of the treatment modality on treatment sequelae, QOL, local control, and survival.

The participating CIRT centers will have considerable heterogeneity regarding patient selection, treatment prescription, total dose per fraction, target and OAR contouring, OAR constraints, and use of RBE dose calculation models. Therefore, CIRT data will be analyzed separately for each institution. In the unlikely event of marked differences in outcomes among these centers, sub-analyses will be performed, and we will consider publishing data for each (deidentified) institution separately.

## 3. Discussion

Although the earliest studies of CIRT were initiated at Lawrence Berkeley National Laboratories in the 1980s, heavy ion therapy has not been used clinically in the US since the Bevalac particle accelerator was decommissioned in 1993. Use of CIRT, however, has increased in Asia and Europe, where 14 centers currently offer this specialized high-LET radiotherapy. In November 2019, Mayo Clinic announced a partnership with Hitachi to develop the first CIRT center in the US, with a single fixed-beam, multi-ion room. The CIRT center will be built along a two-gantry proton beam system in Jacksonville, Florida, and is expected to treat patients with CIRT in 2027. To enhance partnerships and collaborations with other CIRT centers and to begin initial prospective research in the field, Mayo Clinic has led the development of this multicenter clinical trial to evaluate the use of CIRT for sarcoma involving the pelvic bones—one of the least controversial sites for CIRT. The PROSPER study includes acknowledgement of CIRT as an acceptable treatment modality by NCCN experts.

The first and foremost goal of the PROSPER study is to establish a multi-institutional, tricontinental collaboration to enroll patients in a prospective study evaluating the use of CIRT. We expect that this study will provide a basis for several other similar studies that could build on the collaborative network established from this study. The PROSPER study is also expected to provide benchmark measurements of functional QOL, treatment sequelae, and local control rates after surgical treatment, PT, and CIRT. The design is not free from potential biases, especially considering the unique patient population to be enrolled in treatment arm 1. These patients in Europe and Asia have potential differences in health care access, baseline characteristics, and epidemiologic risk factors. However, the benchmark measurements yielded from the PROSPER study could be used for designing the end points of future large Phase III studies. 

## 4. Conclusions

The PROSPER study is an opportunity to initiate clinical research of CIRT in the US, despite the current lack of a clinical CIRT center. We expect to not only gather evidence regarding the advantages of CIRT for the treatment of pelvic sarcomas of the bone but to also learn about CIRT from our partner centers. In particular, we look forward to understanding the intricacies of CIRT treatment planning from our colleagues in the CIRT centers participating in the study and from the centralized database of DICOM plans as Mayo Clinic in Florida prepares to treat its first patient with CIRT in 2027. 

## Figures and Tables

**Figure 1 cancers-15-01660-f001:**
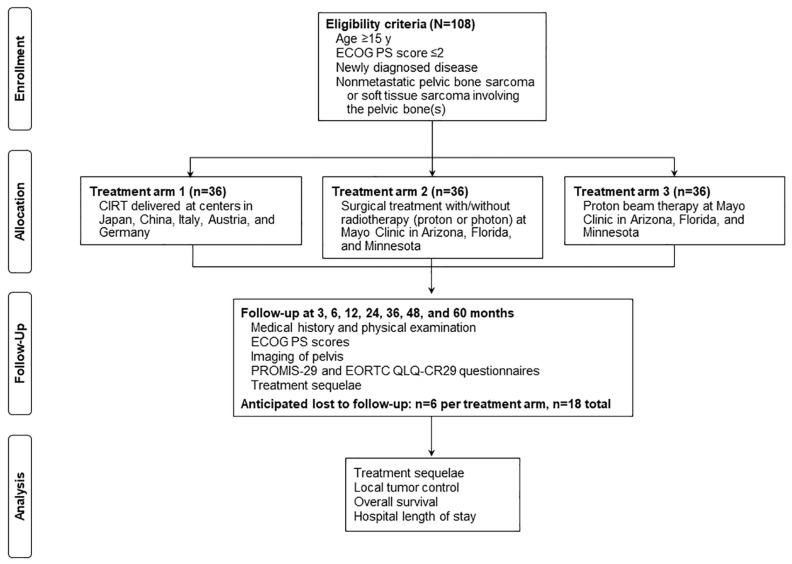
Treatment arms. CIRT indicates carbon ion radiotherapy; ECOG PS, Eastern Cooperative Oncology Group Performance Status; EORTC, European Organisation for Research and Treatment of Cancer; PROMIS-29, Patient-Reported Outcomes Measurement Information System-29; QLQ-CR29, colorectal cancer–specific quality of life.

## Data Availability

The data presented in this study are available within the article.

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
