# Peer review of "Pragmatic, Prospective Comparative Effectiveness Trial of Carbon Ion Therapy, Surgery, and Proton Therapy for the Management of Pelvic Sarcomas (Soft Tissue/Bone) Involving the Bone: The PROSPER Study Rationale and Design"

_cancers, 2023, doi:10.3390/cancers15061660_

Round 1

Reviewer 1 Report

This manuscript outlines the PROSPER protocol, an exciting trial comparing carbon ions, surgery, and protons for pelvic sarcomas in a non-randomized fashion. Overall, I find the manuscript to be well written, with a concise yet descriptive analysis of the background rationale for its utility, methods of enrollment and treatment, and endpoints that will be analyzed in follow-up. While there is limited information provided on treatment specifics, the authors provide discussion that there may be institutional variability on volumes, dose-fractionation, and particle-specific planning aspects, due to the generous eligibility criteria they allowed in order to encourage collaboration.

The authors acknowledge that administrative challenges preclude randomization between surgery and radiation. However, there is a possibility that the differences between patients in each arm will be driven by more than oncologic differences. The European and Asian patient population in Arm 1 will likely be different than those of Arms 2 and 3 in terms of access to healthcare, baseline clinical characteristics, epidemiologic risk factors, etc. Although it is an inevitable limitation of the trial, it is worth discussing. Similarly, subset analyses will be conducted on sacral vs nonsarcal chordomas. Will there be any additional stratifications?

Additional comments are listed below.

Lines 99 and 120

Provide references for alpha/beta of bone tumors and radioresistance of their cancer stem cells.

Line 100

The authors suggest 70 Gy exceeds tolerance of pelvic tissues such a rectum. However, we typically treat to doses higher than 70 Gy when treating prostate cancer with conventional fractionation. Consider re-phrasing or elaborating.

Line 304

The authors mention that the primary outcome PROMIS-29 functional scores will be compared between each arm using 1-sided, 2-sample t tests. They then discuss that they will compare secondary endpoint local control outcomes specifically between Arms 1 and 3 using 1-sided noninferiority tests. Will comparisons be made on whether there are differences between Arm 2 and the other arms for local control?

Author Response

The authors acknowledge that administrative challenges preclude randomization between surgery and radiation. However, there is a possibility that the differences between patients in each arm will be driven by more than oncologic differences. The European and Asian patient population in Arm 1 will likely be different than those of Arms 2 and 3 in terms of access to healthcare, baseline clinical characteristics, epidemiologic risk factors, etc. Although it is an inevitable limitation of the trial, it is worth discussing. Similarly, subset analyses will be conducted on sacral vs nonsarcal chordomas. Will there be any additional stratifications?

Response: We appreciate the insightful comments provided by the reviewer. Additional stratifications may be implemented, but these are undetermined at this time. This is described in the manuscript with the following sentence: “Analysis will be completed for all patients; however, because of the possibility of heterogeneous outcomes resulting from differences in histopathologic findings, analyses of local control will be stratified according to histopathologic characteristics.”

Additional comments are listed below.

Lines 99 and 120

Provide references for alpha/beta of bone tumors and radioresistance of their cancer stem cells.

Response: This is a good point. As suggested, we have inserted the following reference: Elsasser T, Kramer M, Scholz M. Accuracy of the local effect model for the prediction of biologic effects of carbon ion beams in vitro and in vivo. Int J Radiat Oncol Biol Phys. 2008;71:866–872. doi: 10.1016/j.ijrobp.2008.02.037.

Line 100

The authors suggest 70 Gy exceeds tolerance of pelvic tissues such a rectum. However, we typically treat to doses higher than 70 Gy when treating prostate cancer with conventional fractionation. Consider re-phrasing or elaborating.

Response: The reviewer is correct. Therefore, we have modified this sentence to reference the small bowel and lumbosacral plexus, rather than the rectum.

Line 304

The authors mention that the primary outcome PROMIS-29 functional scores will be compared between each arm using 1-sided, 2-sample testsThey then discuss that they will compare secondary endpoint local control outcomes specifically between Arms 1 and 3 using 1-sided noninferiority tests. Will comparisons be made on whether there are differences between Arm 2 and the other arms for local control?

Response: Thank you for your comment. These comparisons will be performed as part of exploratory analyses. The manuscript has been updated to improve clarity regarding these comparisons.

Reviewer 2 Report

The study protocol presented concerns the rationale and the design of the PROSPER Study. This study is a multicontinental, non-randomized, prospective, and 3-arm practical trial. The goals are to evaluate functional outcomes and treatment sequelae in the treatment of sarcoma with surgery, proton therapy, and carbon therapy. This kind of study is very challenging to develop, and it will help to improve the therapeutic modalities of sarcoma by increasing the survival of patients and their quality of life, with a possible change in standard care. 

The design of the study is adapted, especially regarding the complexity of the recruitment in multiple centers and the possibility of comparing the results between themself.

Author Response

Dear Ms Danciu,

We appreciate the opportunity to submit our revised manuscript titled “Pragmatic, Prospective Comparative Effectiveness Trial of Carbon Ion Therapy, Surgery, and Proton Therapy for the Management of Pelvic Sarcomas (Soft Tissue/Bone) Involving the Bone: the PROSPER Study Rationale and Design” (MS ID: cancers-2213890).

We have addressed both the editor and reviewer comments. Our point-by-point responses to the editor comments are below.

Editor comments.

  1. The authorship should be decided based on ICMJE criteria Could you please double-check and confirm the authorship, sign the attached document (handwriting), and send it back to us?

Response: We have double-checked the author list and have included the signed authorship form.

2. Please add the Conclusions section with 1 or 2 paragraphs. We suggest to move the last paragraph of the Discussions to the Conclusion section.

Response: A separate Conclusions section was added after the Discussion.

3. Author Contributions Section, please list every author and every author's detailed contributions during revisions.
Response: Detailed author contributions were added for all authors.

4. Please also add the date for the approval in the Institutional Review Board Statement.

Response: The specific date of approval by our institutional review board has been added.

5. Please rephrase the Data Availability Statement from "Not applicable." to "The data presented in this study is available within the article."

Response: We have revised the Data Availability Statement as suggested.

6. The institutional emails of the following authors as we have noticed that these emails are not institutional ones: Mx. Qing Zhang - [email protected], Mx. Cihang Bao - [email protected]

Response: These email addresses have been revised as requested.

  1. Titles (MD, PhD, Dr., etc.) are not allowed in the Acknowledgments section, please remove all the titles.

Response: This has been corrected.

Sincerely,

Bradford Hoppe MD, MPH
